# Comprehensive assessment of miniature CRISPR-Cas12f nucleases for gene disruption

Changchang Xin[1,2], Jianhang Yin[1,2], Shaopeng Yuan[1], Liqiong Ou[1], Mengzhu Liu[1], Weiwei Zhang[1] & Jiazhi Hu [1] ✉

Because of their small size, the recently developed CRISPR-Cas12f nucleases can be effectively packaged into adeno-associated viruses for gene therapy. However, a systematic evaluation of the editing outcomes of CRISPR-Cas12f is lacking. In this study, we apply a high-throughput sequencing method to comprehensively assess the editing efficiency, specificity, and safety of four Cas12f proteins in parallel with that of Cas9 and two Cas12a proteins at multiple genomic sites. Cas12f nucleases achieve robust cleavage at most of the tested sites and mainly produce deletional fragments. In contrast, Cas9 and Cas12a show relatively higher editing efficiency at the vast majority of the tested sites. However, the off-target hotspots identified in the Cas9- and Cas12a-edited cells are negligibly detected in the Cas12f-edited cells. Moreover, compared to Cas9 and Cas12a nucleases, Cas12f nucleases reduce the levels of chromosomal translocations, large deletions, and integrated vectors by 2- to 3-fold. Therefore, our findings confirm the editing capacity of Cas12f and reveal the ability of this nuclease family to preserve genome integrity during genome editing.

The clustered regularly interspaced short palindromic repeats (CRISPR)-CRISPR-associated protein (Cas) system in bacteria has been engineered into a robust genome-editing tool that can be used to manipulate the mammalian genome efficiently. The earliest Cas protein to be applied to mammalian gene editing was *Streptococcus pyogenes* Cas9 (*Sp*Cas9) in the type II CRISPR-Cas system[1,2]. Then, type V-A Cas12a, which mainly includes *Acidaminococcus* sp. Cas12a (*As*Cas12a) and *Lachnospiraceae bacterium* Cas12a (*Lb*Cas12a), was demonstrated to be powerful for gene modulation[3–5]. Cas9 and Cas12a are the most prevalent gene-editing tools and show great potential for clinical use[6–11]. However, because they are large, neither of these proteins can be efficiently packaged into a single adeno-associated viral (AAV) particle, limiting the broad application of Cas9 and Cas12a systems in vivo[12–14]. Hence, new miniature CRISPR-Cas systems have been recently developed to accommodate the payload size (<4.7 kb) needed for AAV delivery[15,16]. For example, the Class II type V-E nuclease *Planctomycetes* Cas12e (*Plm*Cas12e, also known as *Plm*CasX), a 978-amino acid (aa) protein, has been reported to functionally edit the human genome[17–19].

Moreover, the most recently reported Class II type V-F Cas12f proteins, including *Un1*Cas12f1 from an uncultured archaeon and previously named Cas14a1[20–22], CasMINI (a protein engineered from *Un1*Cas12f1), and *Acidibacillus sulfuroxidans* Cas12f1 (*As*Cas12f1), consisting of 529 aa, 529 aa, and 422 aa, respectively, have shown robust mammalian cell-editing activity maintaining a small protein size[23,24].

Cas12e recognizes a 5′-TTCN protospacer adjacent motif (PAM) and generates two broken sticky ends by predominantly cleaving DNA at two distinct positions -10 nucleotides (nt) apart in a nontarget strand (NTS) and target strand (TS)[17]. This cleavage pattern is similar to that of Cas12a but different from that of Cas9. *Un1*Cas12f1 cleaves double-stranded DNA with a 5′-TTTR PAM and introduces DNA breaks in a staggered cutting pattern, and the overhang can be as long as 10 base pairs (bp)[22]. *As*Cas12f1 has been shown to predominantly cleave DNA at the 3rd nt downstream of a spacer in the TS and simultaneously generate two breaks in the NTS, with one located at the 12th nt and the other located at approximately the 23rd–27th nt downstream of a PAM, producing 5′-overhangs of ~11-bp[24]. The emergence of these

[1]The MOE Key Laboratory of Cell Proliferation and Differentiation, Genome Editing Research Center, School of Life Sciences, Peking-Tsinghua Center for Life Sciences, Peking University, 100871 Beijing, China. [2]These authors contributed equally: Changchang Xin, Jianhang Yin. ✉e-mail: hujz@pku.edu.cn

miniature CRISPR-Cas toolboxes has broadened the target range for editing the human genome and provides a convenient choice for gene delivery in the clinic.

However, in addition to inducing gene disruptions at a target site, CRISPR-Cas can cause unintended off-target cleavage at imperfectly matched loci[25–28]. Moreover, it can lead to unwanted structural variations in the chromosome, including chromosomal translocations, large deletions, and integration of exogenous DNA[29–36]. These byproducts severely threaten genome integrity and are associated with oncogenesis, raising great concern over the safety of genome editing. Comprehensive assessments of these newly developed miniature Cas12 nucleases on the specificity and genomic structural variations generated during genome editing are lacking, which has restricted the further optimization of miniature Cas enzyme-guided strategies in clinical gene-editing applications[37].

In this study, we assess the activity and safety of the above-mentioned Cas nucleases in depth. Here, we present a full spectrum of editing outcomes, including off-target mutations and structural variations, induced by Cas12f, which are evaluated in parallel with Cas9 and Cas12a. We conclude that Cas12f can induce efficient cleavage at some genomic sites with fewer off-target cleavage events and structural variations. Among the Cas12f types tested, CasMINI shows the most consistent editing ability and specificity at most of the tested genomic sites.

## Results

### Assessment of the editing ability of Cas12f nucleases as determined by EGFP silencing assay

To compare the editing ability of various CRISPR-Cas12 nucleases, we constructed a destabilized-enhanced green fluorescent protein (EGFP)

reporter system by integrating the EGFP gene into the *AAVS1* safe harbor locus in HEK293T cells (Fig. 1a). We then cloned *Un1*Cas12f1, CasMINI, *As*Cas12f1, and *Plm*Cas12e as well as *Sp*Cas9, *As*Cas12a, and *Lb*Cas12a into the same plasmid backbone and coexpressed these plasmids with a puromycin resistance gene fused via a P2A self-cleavage peptide (Fig. 1b, c). Notably, optimized single guide (sg) RNA version 4.1 was used for *Un1*Cas12f1 because it had been previously reported to show effective editing[20]. We also introduced this engineered sgRNA_*ge4.1* into CasMINI to generate CasMINI_*ge4.1*. To test the editing efficiency, we designed two target sites for each CRISPR-Cas nuclease (Supplementary Fig. 1a), in which the gRNAs recognized the EGFP locus at similar loci but with offsets to satisfy differences in PAM specificity (Fig. 1b). The cells were selected with puromycin for 3 days and then assayed for GFP disruption by flow cytometry 5 and 10 days post-transfection (Fig. 1c). We used the proportion of GFP-negative cells to estimate the gene disruption rates of each nuclease, although the former may have been lower than the latter measure (Supplementary Fig. 1b, c).

Complete degradation of GFP requires a long time, and we indeed detected an obvious increase in the GFP-negative cell population 10 days post-edit compared to that observed 5 days post-edit (Supplementary Fig. 1b). Then, we used the proportion of GFP-negative cells measured 10 days post-transfection to make a final assessment. All the tested editing nucleases induced varying degrees of gene expression disruption, with Cas9 and two Cas12a enzymes exhibiting the highest editing efficiency at both target sites (>42.5%) (Fig. 1d and Supplementary Table 1). *Un1*Cas12f1_*ge4.1* and CasMINI_*ge4.1* showed relatively lower but useful levels of editing efficiency at the two target sites, ranging from 18.3 to 26.2%. Notably, CasMINI exhibited a high efficiency at

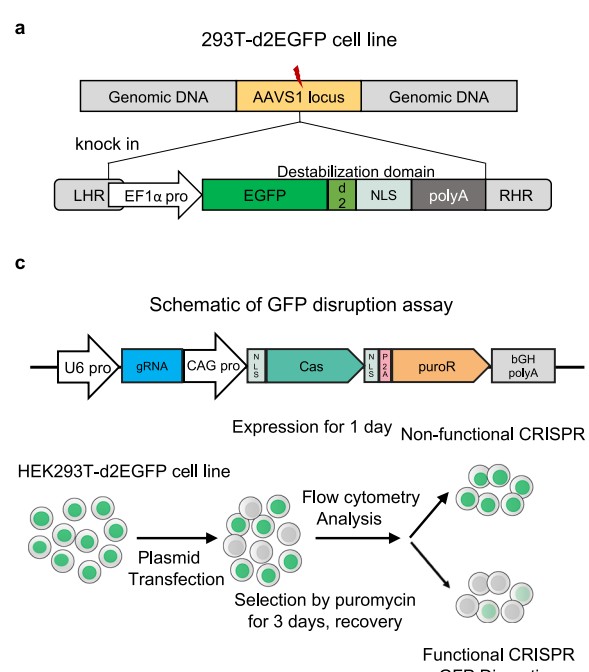

**a**
293T-d2EGFP cell line

**c**
Schematic of GFP disruption assay

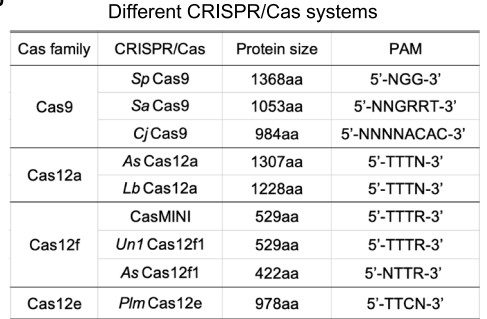

**b** Different CRISPR/Cas systems

| Cas family | CRISPR/Cas | Protein size | PAM |
|---|---|---|---|
| Cas9 | *Sp* Cas9 | 1368aa | 5'-NGG-3' |
|  | *Sa* Cas9 | 1053aa | 5'-NNGRRT-3' |
|  | *Cj* Cas9 | 984aa | 5'-NNNNACAC-3' |
| Cas12a | *As* Cas12a | 1307aa | 5'-TTTN-3' |
|  | *Lb* Cas12a | 1228aa | 5'-TTTN-3' |
| Cas12f | CasMINI | 529aa | 5'-TTTR-3' |
|  | *Un1* Cas12f1 | 529aa | 5'-TTTR-3' |
|  | *As* Cas12f1 | 422aa | 5'-NTTR-3' |
| Cas12e | *Plm* Cas12e | 978aa | 5'-TTCN-3' |

**d**

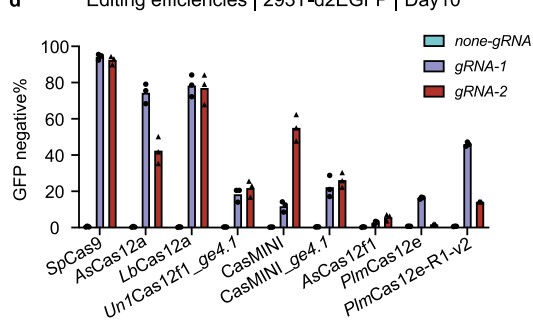

Editing efficiencies | 293T-d2EGFP | Day10

**Fig. 1 | Cas12f nucleases efficiently disrupted EGFP gene expression in vivo.** **a** Schematics showing the 293T-d2EGFP cell lines. Lightning represents the position of *Sp*Cas9-induced double-strand break (DSB). A gene target plasmid to cut the *AAVS1* site and an integration donor plasmid are used to achieve the site-specific exogenous d2EGFP gene insertion. LHR and RHR represent left or right homo-arm, respectively. "pro" represents promoter. **b** Summary of different CRISPR-Cas systems. **c** Schematic showing the editing efficiency detection assay with the HEK293T-d2EGFP cell lines. "pro" represents promoter. "puroR" represents the puromycin resistance gene. "NLS"

represents nuclear localization sequence. The plasmids carrying Cas nuclease and single guide (sg) RNA that could target the EGFP gene are transfected to d2EGFP cells. After the protein is expressed 1 day, add puromycin to select the positive transfection cells and do the flow cytometry analysis. **d** The GFP disruption efficiency of different Cas nucleases at the indicated target sites. The GFP disruption proportion is referred to as the number of GFP-negative cells relative to the total number of cells. "ng" represents nontargeting guide RNA. "g1" and "g2" indicate site 1 and site 2, respectively (*N* = 3, mean ± SD from three biological replicates).

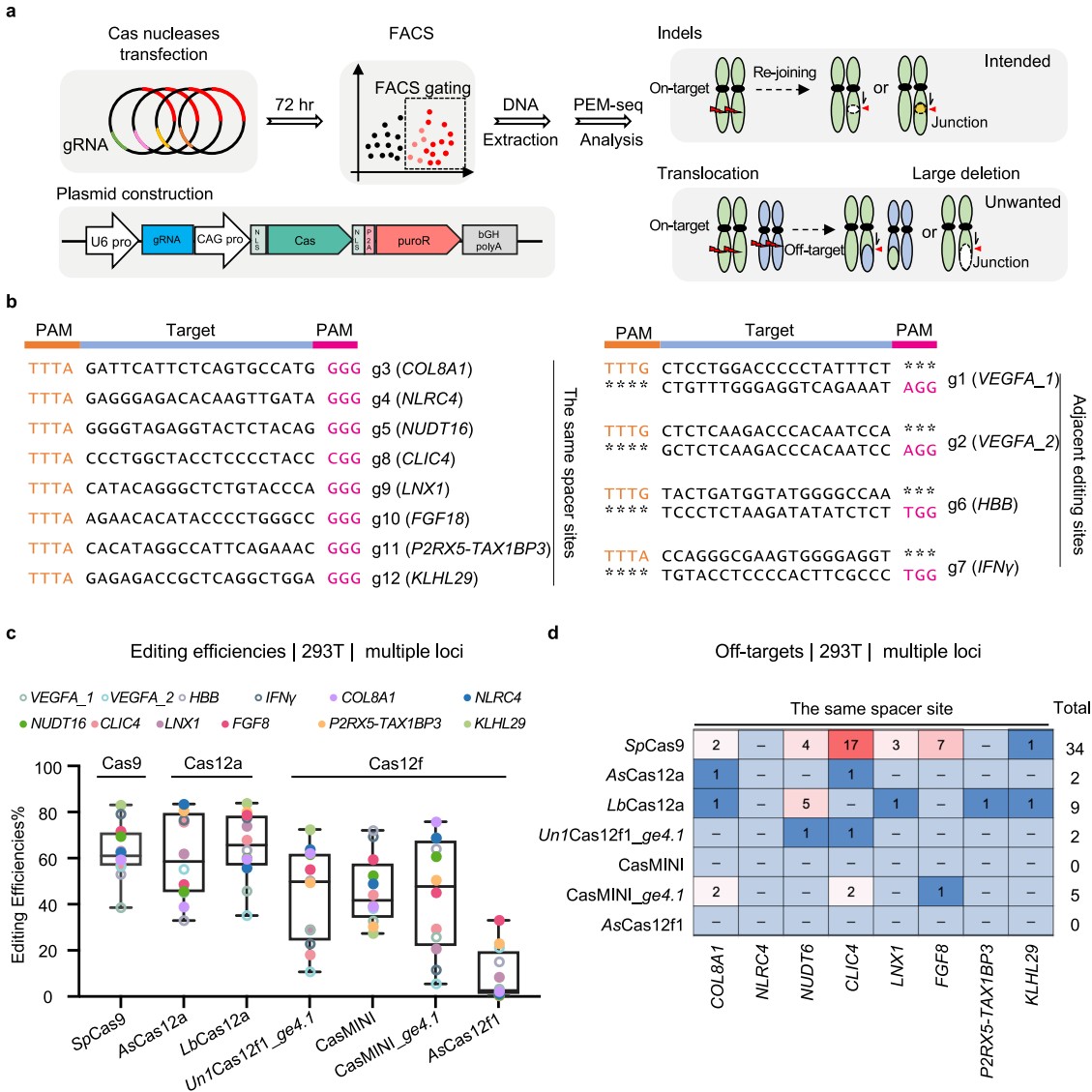

**Fig. 2 | Evaluation of the global editing outcomes of Cas12f nucleases in parallel with Cas9 and Cas12a as determined by PEM-seq. a** Schematic diagram showing Cas nuclease evaluation by PEM-seq. The plasmids carrying both Cas nuclease-mCherry and single guide (sg) RNA were transfected into HEK293T cells, and the successfully transfected cells were sorted via FACS 72 h post-transfection followed by PEM-seq construction. PEM-seq can simultaneously detect and quantify small indels, large deletions, and chromosomal translocations with off-targets or general double-strand breaks (DSBs). **b** Detailed guide (g) RNA sequence information of the 12 target sites with PAM sequences for Cas12 and Cas9 in orange or fuchsia, respectively. These target sites were categorized into adjacent editing sites with the second target site (55.1%) and also cleaved the first site with an different gRNA sequences and same-spacer sites with consistent gRNA sequences. **c** Editing efficiency of the Cas nucleases at the indicated 12 loci as detected by PEM-seq. Editing efficiency indicates the total percentage of insertions, deletions, and translocations. Values from minimum to maximum are shown by the whiskers, and the bounds of the box indicate the first and third quartile (*N* = 12). The vertical line through the box is the median. Source data are provided as a Source Data file. **d** Number of off-target editing at the same-spacer sites as detected by PEM-seq methods and identified through the PEM-Q pipeline. "-" means there was no detected off-target site. The indicated locus and Cas nuclease information are marked.

efficiency of 11.8%. Notably, CasMINI_*ge4.1* did not show better activity than CasMINI. Moreover, *As*Cas12f1 and *Plm*Cas12e showed only slight GFP silencing at both sites (<5.7% and <16.3%, respectively) (Fig. 1d). Recently, an optimized Cas12e variant, *Plm*Cas12e-R1-v2, has been reported to be able to enhance editing efficacy[18]. We found that *Plm*Cas12e-R1-v2 indeed improved editing efficiencies at both target sites than *Plm*Cas12e (46.0% vs. 16.3% and 14.3% vs. 1.8%, respectively) (Fig. 1d and Supplementary Fig. 1b). These results implied that most of the Cas12f nucleases and the optimized Cas12e can be used for gene editing and encouraged us to perform additional experiments to explore the details of their editing activity.

**CRISPR-Cas12f nucleases show high fidelity during gene editing**

We next employed a previously developed primer-extension-mediated sequencing (PEM-seq) method to profile the editing outcomes of Cas12f nucleases, assessing both intended small insertions/deletions (indels) and unwanted chromosomal rearrangements such as large deletions, off-target translocations, and general translocations (Fig. 2a)[30,38]. We designed 12 target sites within or adjacent to the human *VEGFA*, *HBB*, *IFNγ*, *COL8A1*, *NLRC4*, *NUDT16*, *CLIC4*, *LNX1*, *FGF18*, *P2RX5-TAX1BP3*, and *KLHL29* genes and used the nine abovementioned CRISPR-Cas nucleases to target these sites in HEK293T cells. Eight of the 12 target sites had the same spacer sequence with a TTTA PAM for Cas12 and an NGG PAM for Cas9. The other four gRNAs were designed to target adjacent editing sites within a narrow range of locations to

satisfy differences in PAM specificity (Fig. 2b and Supplementary Table 2). The cells were collected by fluorescence-activated cell sorting (FACS) based on mCherry fluorescence 72 h post-transfection. Genomic DNA was then isolated to prepare PEM-seq libraries (Fig. 2a and Supplementary Fig. 2a).

SpCas9, AsCas12a, and LbCas12a showed comparably high levels of editing efficiency, and Un1Cas12f1_ge4.1, CasMINI_ge4.1, and Cas-MINI exhibited robust cleavage at the 12 tested sites (Fig. 2c). In contrast, AsCas12f1 effectively cleaved only 5 of 12 target sites (Fig. 2c and Supplementary Data 1), in line with the findings of the eGFP-silencing assay. Unexpectedly, PlmCas12e showed very limited or even undetectable cleavage at all target sites; despite the optimized PlmCas12e-R1-v2 improved editing efficacies at some tested sites, only three sites showed editing efficiencies over 5% (Supplementary Fig. 2b and Supplementary Data 1); therefore, we excluded both PlmCas12e and PlmCas12e-R1-v2 from further analyses. In addition to its high-editing ability, SpCas9 showed robust off-target activity at most of the editing sites, with the number of identified off-target sites being 0 for three sites and 1–17 for the other nine sites (Fig. 2d and Supplementary Fig. 2c and Supplementary Data 2). In contrast, AsCas12a and LbCas12a showed lower off-target activity, and even fewer off-target sites were identified for the Cas12f family nucleases at both the same spacer sites and adjacent editing sites (Fig. 2d and Supplementary Fig. 2c, d). Notably, no off-target site was identified for CasMINI or AsCas12f1, although CasMINI showed editing efficiency at these tested sites comparable to that of the other CRISPR-Cas enzymes. Collectively, these results indicate that Cas12f family nucleases, especially CasMINI, show higher specificity than Cas9 or Cas12a.

### Cas12f nucleases tend to induce small deletions during gene editing

We next sought to investigate repair outcomes during gene targeting by CRISPR-Cas12f. Strikingly, the vast majority of the editing events were deletions by Cas12a and Cas12f, with percentages greater than 92%, from the lowest for LbCas12a (92.4%) to the highest for AsCas12f1 (96.4%), and all were significantly higher than the percentage of deletions induced by SpCas9 (69.9%) (Fig. 3a)[20,24]. The preponderance of deletions by Cas12a or Cas12f may have been a result of the staggered cleaved DNA ends being processed by endogenous DNA nucleases promoting deletions not insertions. In this context, we found that the Cas12a- or Cas12f-edited products were highly enriched with small deletions (<100 bp) with most small deletions with lengths correlated to the distance between the two staggered cleavage sites of Cas12a or Cas12f (Fig. 3b, c). Specifically, SpCas9 showed a higher proportion of 1–2-bp deletion fragments, while AsCas12a- and LbCas12a-edited products were largely 3–7 bp deletion fragments, and the action of the four Cas12f enzymes led to 2–11-bp deletion products, as exemplified by the highly edited FGF18 target sites (Fig. 3c, Supplementary Figs. 3a and 4a).

We further explored the distribution patterns of deletions induced by these CRISPR-Cas enzymes. The distribution profiles of the Cas12a and Cas12f family enzymes resembled the profile of SpCas9, showing similar distribution patterns of gross deletions (Supplementary Fig. 4b). However, compared to SpCas9, both Cas12a and Cas12f family enzymes showed higher percentages of deletions from 20- to 60-bp (Fig. 3d and Supplementary Data 3). Notably, the engineered sgRNA_ge4.1 led to an increase in deletion length, as Un1Cas12f1_ge4.1 and CasMINI_ge4.1 induced more deletions from 20-to 60-bp than CasMINI (32.9% and 31.3% vs. 21.3%; Fig. 3d). In addition, all the tested CRISPR-Cas enzymes generated very few deletion fragments of 60 to 100 bp (Fig. 3d). The percentage of deleterious large deletions induced by SpCas9 was 1.92%. Cas12a generated large deletions to an extent similar to SpCas9, with 1.40% and 2.29% large deletions produced by AsCas12a and LbCas12a, respectively. In contrast, 1.23% and 1.27% of the deletion fragments produced by Un1Cas12f1_ge4.1 and CasMINI_ge4.1,

respectively, were large deletions, with and the percentages even smaller, at 0.93% and 0.50%, for CasMINI and AsCas12f1, respectively (Fig. 3e). Therefore, the Cas12f family nucleases, especially CasMINI and AsCas12f1, induce only a limited number of large deletions.

### Cas12f nuclease activity leads to fewer integrated vectors than those generated by SpCas9 during genome editing

Compared to those of deletions, the percentages of insertions produced by Cas12a and Cas12f family nucleases were lower, falling to 2.4–5.9% from 26.6% at the 12 target sites for SpCas9 (Fig. 4a). Consequently, more insertions of all lengths, from 1 bp, 2–25 bp to 25–40 bp and even larger fragments, were observed after SpCas9 editing (Fig. 4b and Supplementary Data 4). We noticed that the insertions in the KLHL29, COL8A1 and CLIC4 target loci induced by SpCas9 were 83.1%, 48.4%, and 43.7%, respectively, higher than the average levels of insertions (Fig. 4a and Supplementary Fig. 5a). These exceptionally frequent insertions were results of 1-bp insertions that were identical to the fourth nt upstream of NGG, which occupied 93.9%, 84.1%, and 78.9% of the total insertion events in KLHL29, COL8A1, and CLIC4, respectively (Supplementary Fig. 5a, b), in line with previous findings[39,40]. In addition to the abovementioned 1-bp insertions, the Cas12a and Cas12f family nucleases produced fewer insertions of 40- and 60-bp and more insertions of 2- and 25-bp, as exemplified by the length distribution at the HBB site (Fig. 4c and Supplementary Fig. 5c). The enrichment of insertions in the 2- to 25-bp range due to Cas12a and Cas12f editing might be explained by gaps filled in with generated sticky ends before rejoining (Supplementary Fig. 5d), while the insertions in the 40- to 60-bp might involve vector integration[31].

Next, we extracted the inserted DNA sequences from PEM-seq libraries and aligned them to the Cas nuclease-expressing vectors at all tested loci and found hundreds to thousands of distinct inserted sequences originating from the transfected vectors. The frequency of vector insertion by SpCas9 was ~5.2 thousand per 100 thousand on-target indels. The Cas12a and Cas12f family enzymes exhibited a slight decrease in vector integration level, while CasMINI and AsCas12f1 showed a more significant decrease in vector integration (Fig. 4d and Supplementary Table 3). We then mapped the integrated vector fragments across the respective plasmids and found that the inserted vector fragments had been distributed across the plasmid backbone, with accumulation at the AAV inverted terminal repeat region for all the nucleases (Fig. 4e), in line with previous reports[31,41]. In conclusion, the Cas12a and Cas12f family nucleases induce vector integration at a lower rate than SpCas9.

### Cas12f induces substantial genomic structural variations

Genomic structural variations, including the abovementioned large deletions and chromosomal translocations, are the most deleterious editing products generated during CRISPR–Cas9 genome editing[32,41]. In this context, we found that ~3.55% of all edits in SpCas9-edited cells were chromosomal translocations. The percentages of translocations were reduced to 1.42% and 1.77% for AsCas12a and LbCas12a, respectively. Un1Cas12f1_ge4.1 and CasMINI_ge4.1 induced translocations at a level similar to that induced by Cas12a, with percentages of 1.58% and 1.32%, respectively. However, the percentage of translocations induced by either CasMINI or AsCas12f1 was lower, at 1.17% (Fig. 5a, b). Notably, the abundance of translocated fragments was site-specific and varied among different loci (Fig. 5a). In addition, the identified translocations were widely distributed across the whole genome, with obvious enrichment at off-target sites for all the CRISPR-Cas enzymes, as exemplified by the translocations at the FGF8 target site (Fig. 5b). To comprehensively evaluate the effects of these Cas nucleases, we calculated the editing safety score of these enzymes by combining their off-target activity with the extent of structural variations they induced at effectively edited sites and aligned the score on the basis of the editing efficiency score. The distribution profile of the resulting two-

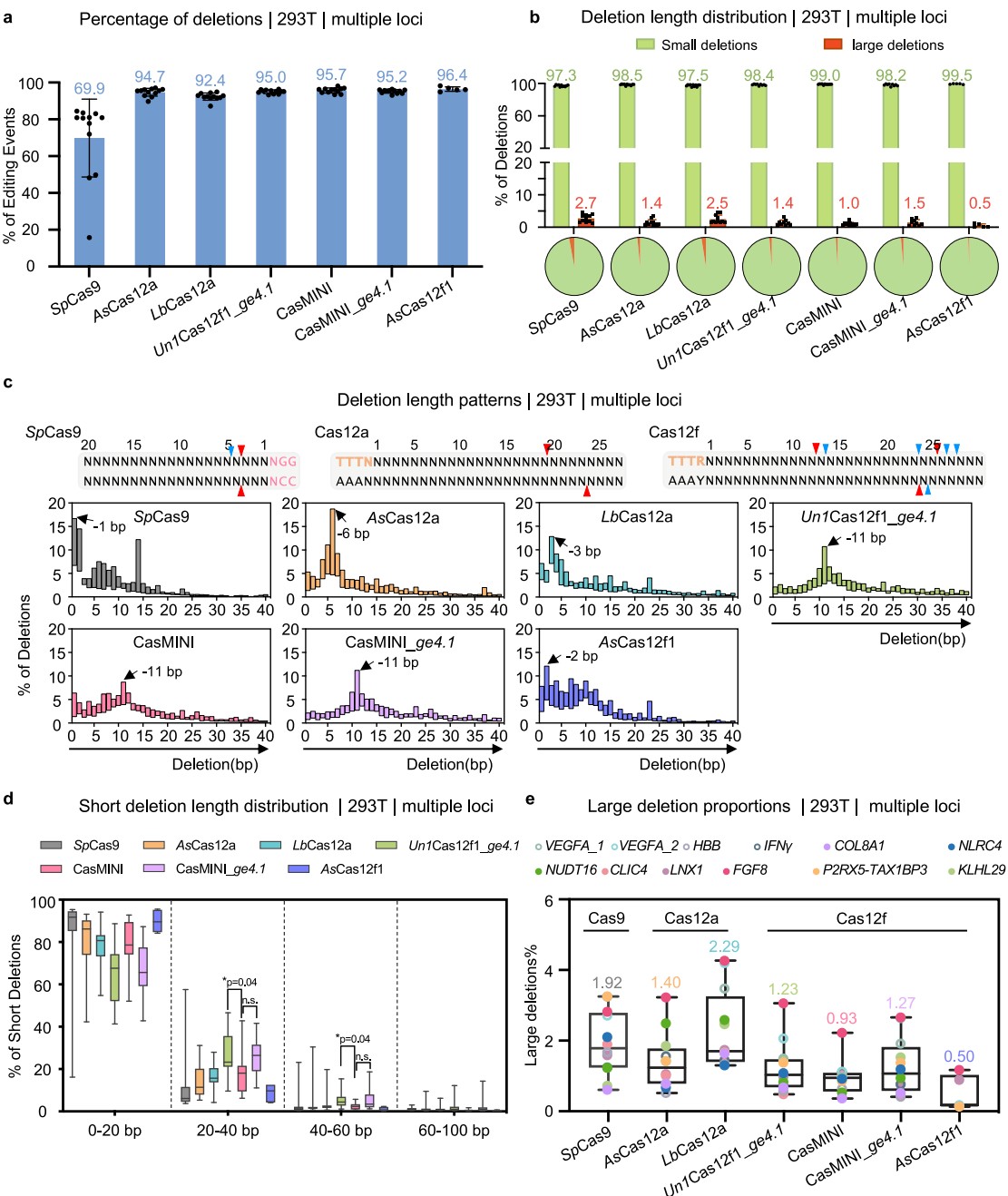

**Fig. 3 | Compared to Cas9, the Cas12a and Cas12f nucleases generated more deletion fragments. a** Bar chart showing the percentages of deletions for the indicated Cas nucleases as detected by PEM-seq. The numbers above the bars refer to the average percentages of deletion fragments at all sites for which editing activity was observed (For *Sp*Cas9, *As*Cas12a, *Lb*Cas12a, *Un1*Cas12f1_*ge4.1*, CasMINI and CasMINI_*ge4.1*, *n* = 12; for *As*Cas12f1, *n* = 5. Data are presented as mean ± SD.). **b** Above: bar chart showing the percentages of small deletions (≤100 base pairs (bp), in green) and large deletions (>100 bp, in red) at 12 loci as detected by PEM-seq in HEK293T cells. Bottom: pie chart showing the average percentages of indicated deletions at 12 loci (*N* = 12, data are presented as mean ± SD. Notably, for *As*Cas12f1, *n* = 5). **c** Above: the double-strand DNA cleavage patterns induced by Cas9, Cas12a, and Cas12f nucleases. Red and blue arrowheads indicate the major and minor cleavage sites, PAM sequences for Cas12 and Cas9 are in orange or fuchsia, respectively. Bottom: size and positional information of the deletions, within a length of 40 bp, generated by the indicated Cas nucleases at all tested sites. The vertical axis indicates the average ratio refers to the number of deletion fragments with the indicated length to the total number of deletion events. The most

abundant deletion size for all tested nucleases is indicated by the black arrow. **d** The distribution of short deletions with the indicated length for all tested nucleases at 12 loci in HEK293T cells. Short deletions were divided into four lengths: 0–20 bp, 20–40 bp, 40–60 bp, and 60–100 bp, and the vertical axis indicates the number of special deletions to total number of short deletions. Values from minimum to maximum are shown by the whiskers, and the bounds of the box indicate the first and third quartile (*N* = 12, for *As*Cas12f1, *n* = 5). The vertical line through the box is the median. In both 20–40 bp and 40–60 bp deletions, One-way ANOVA with Geisser-Greenhouse correction analysis was performed for the three data sets: *Un1*Cas12f1_*ge4.1*, CasMINI, and CasMINI_*ge4.1*. *p* < 0.1, n.s. not significant. **e** The percentage of large deletions caused by the indicated Cas nucleases on the basis of the total editing events for the indicated 12 loci in HEK293T cells as detected by PEM-seq, with the numbers above the whiskers referring to the average percentages of large deletions at all sites. Values from minimum to maximum are shown by the whiskers, and the bounds of the box indicate the first and third quartile (*N* = 12, for *As*Cas12f1, *n* = 5). The vertical line through the box is the median. Source data are provided as a Source Data file.

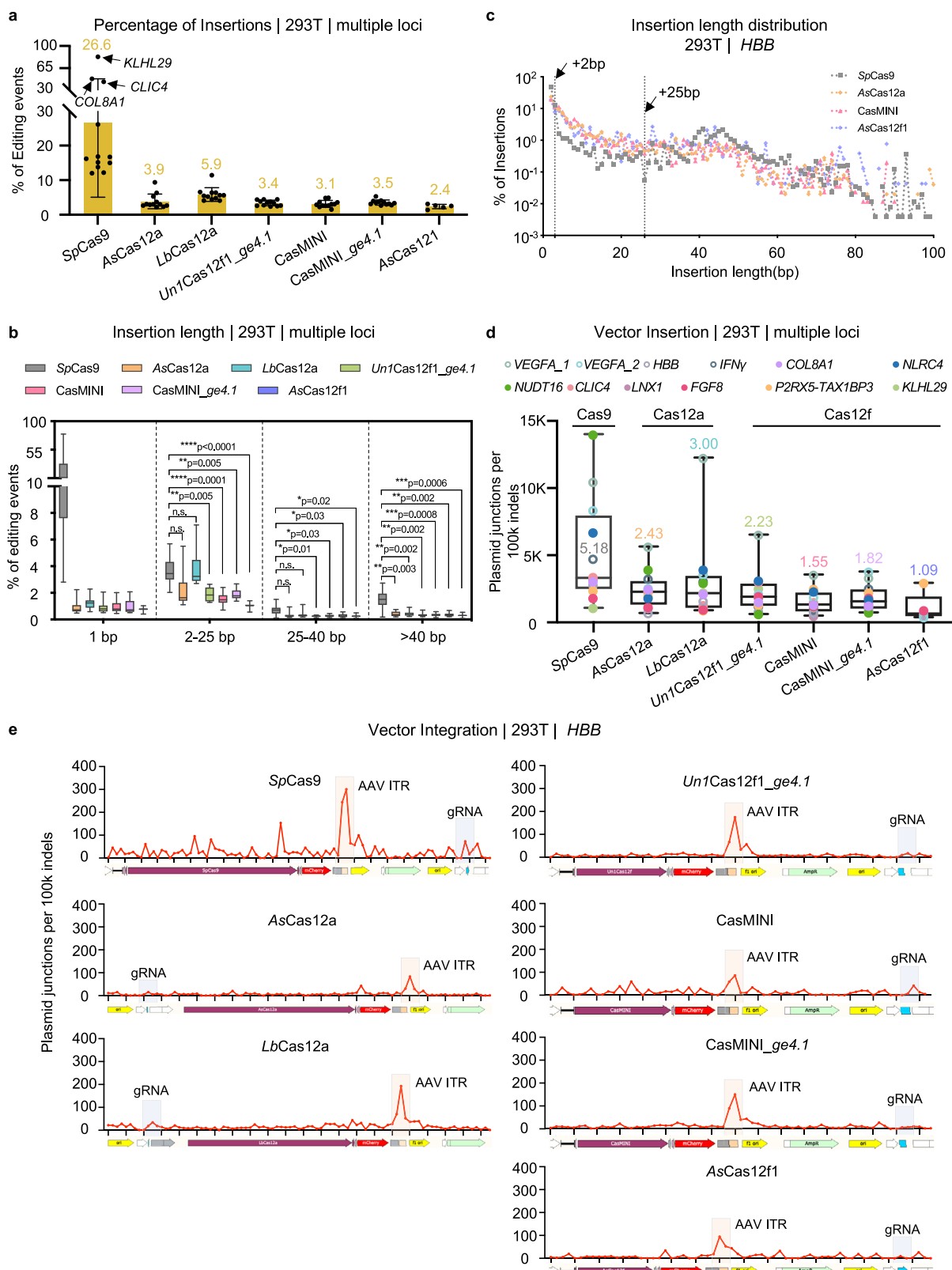

score map indicated that both CasMINI- and *As*Cas12f1-editing led to accurate editing with high safety, while CasMINI editing was more efficient than that of *As*Cas12f1 (Fig. 5c).

## Discussion

Genome editing has been or will be used in therapeutic applications for many genetic disorders[8,42,43]. The accuracy, precision, and especially safety of genome editing have raised grave concerns for the clinical applications of powerful gene-editing tools, including the CRISPR-Cas system. In fact, from the clinical perspective, the safety and deliverability of editing tools have been a longstanding consideration[37,44]. The miniature CRISPR-Cas12f system, with a minimal protein size, ranging from 400 to 700 aa, was developed to overcome application difficulties due to the payload size of the AAV delivery system[20–24]. In this

**Fig. 4 | Cas12f nucleases suppressed vector integration during gene editing.**
**a** Bar chart showing the percentages of insertions for the indicated Cas
nucleases as detected by PEM-seq, with the numbers above the bars referring to
the average percentages of insertions at all sites for which editing activity was
observed (For *Sp*Cas9, *As*Cas12a, *Lb*Cas12a, *Un1*Cas12f1_ge4.1, CasMINI and
CasMINI_ge4.1, n = 12; for *As*Cas12f1, n = 5. Data are presented as mean ± SD). The
*KLHL29*, *COL8A1*, and *CLIC4* loci in the *Sp*Cas9 panel are indicated by black
arrows. **b** The distribution of insertions with the indicated length for all tested
nucleases at 12 loci in HEK293T cells. Insertions were divided into four lengths: 1
base pair (bp), 2–25 bp, 25–40 bp, and >40 bp, and the vertical axis indicates the
number of special insertions to the number of total insertions. Values from
minimum to maximum are shown by the whiskers, and the bounds of the box
indicate the first and third quartile (N = 12, for *As*Cas12f1, n = 5). The vertical line
through the box is the median. In 2–25 bp, 25–40 bp, and >40 bp insertions,
One-way ANOVA with Geisser-Greenhouse correction analysis was performed
for all seven data sets: *Sp*Cas9, *As*Cas12a, *Lb*Cas12a, *Un1*Cas12f1_ge4.1, CasMINI,
CasMINI_ge4.1, and *As*Cas12f1. n.s. not significant, *p ≤ 0.1, **p ≤ 0.01,

***p ≤ 0.001, ****p ≤ 0.0001. **c** The insertion length distribution with the indi-
cated length at the *HBB* locus in HEK293T cells. Total insertion junctions are
plotted on the log scale. The different colors indicate different Cas nucleases.
Black arrows mark 2-bp insertions and 25-bp insertions. **d** Statistical analysis of
vector integration junction numbers per 100k on-target indels for different Cas
nucleases at each of the 12 loci, as detected by PEM-seq cloning from the on-
target region, with the numbers above the whiskers or within the boxes refer-
ring to the average percentages of vector integration numbers at all sites. K
means thousand. Values from minimum to maximum are shown by the whiskers,
and the bounds of the box indicate the first and third quartile (N = 12, for
*As*Cas12f1, n = 5). The vertical line through the box is the median. Source data are
provided as a Source Data file. **e** The distribution of vector cleavage and inte-
gration junctions across the respective plasmids for every 100k indels for the
different Cas nucleases at the *HBB* locus as detected by PEM-seq. K means
thousand. Bin size = 100 bp. The adeno-associated virus (AAV) inverted repeat
(ITR) region and the guide (g) RNA scaffold are highlighted with pale-yellow and
light blue shadows, respectively.

study, we systematically assessed the editing properties of Cas12f
nucleases in parallel with the prevalently used Cas9 and Cas12a
nucleases. Cas12f showed robust cleavage activity in the mammalian
genome but with an efficiency generally lower than that of Cas9 or
Cas12a; moreover, *As*Cas12f1 showed poor editing outcomes at most
target sites (Fig. 2c). *As*Cas12f1 is a thermophilic nuclease, and this
property might prevent *As*Cas12f1 from achieving maximum effec-
tiveness in human cells[24]. Regarding the intended editing products,
including mostly indels, compared to Cas9, both Cas12a and Cas12f
tended to induce more deletional products, which were correlated
with the overhang length generated by the asymmetrical cleavage of
two DNA strands by Cas12 enzymes (Fig. 3a, c). In this context, a Cas12-
based strategy may be useful for editing the genomes in genetic dis-
eases for which specific DNA fragment deletions are desired[45,46]. Cor-
respondingly, the 1-bp insertions that were abundant among *Sp*Cas9-
edited products and the number of integrated vectors were not typical
products of Cas12a or Cas12f editing (Fig. 4b–d and Supplemen-
tary Fig. 5a).

Furthermore, the Cas12f nucleases were characterized by rela-
tively higher specificity and safety than Cas9 and Cas12a nucleases.
The observed higher specificity of the Cas12f nucleases might be due
to the overall lower activity at the on-target sites in comparison to
Cas9 and Cas12a nucleases. However, we also noticed that both
CasMINI and *As*Cas12f1 had undetectable off-target effects at the
tested sites and generated very few large deletions or translocations
in some effectively edited loci (Figs. 2d, 3e, 5a and Supplementary
Fig. 2b), which suggested that the long over-hangs (-11 bp) of cleaved
ends may be also involved in suppressing structural variations by
affecting the DNA repair pathways. Conclusively, in cases where the
Cas12f enzyme can efficiently induce DNA modifications at target
sites, Cas12f nucleases may help maintain genome integrity. In the
context of editing ability, CasMINI is a relatively better choice than
*As*Cas12f1, but the editing efficiency of CasMINI needs to be eval-
uated before use (Fig. 5c). For example, further study with mouse
disease models is needed to validate the in vivo properties of Cas-
MINI. Finally, further optimization is required to further broaden the
target range or improve the editing efficacy of CasMINI as well as
*As*Cas12f1.

## Methods
### Plasmid construction
Different Cas nucleases were cloned into the same pX330 plasmid
vector (Addgene ID 42230) with puromycin resistance gene coex-
pression with the P2A self-cleavage peptide for 293T-d2EGFP assay or
with the mCherry marker gene for cell sorting in the PEM-seq assay.
The DNA sequences of different Cas nucleases and gRNA scaffolds
were displayed in Supplementary Data 5. The gRNA was cloned into the

same vector with a U6 promoter. All the gRNA sequences are shown in
Supplementary Table 2.

### Cell culture and plasmid transfection
HEK293T cells (a gift from Dr. Frederick Alt Lab, Harvard Medical
School) were cultured in Dulbecco's modified Eagle's medium (Corn-
ing) with 10% Fetal Bovine Serum (ExCell Bio), Penicillin–Streptomycin
(Corning), and L-Glutamine (Corning) at 37 °C with 5% $CO_2$.
HEK293T cells cultured in 6-cm dishes were transfected with 6 μg of
pX330-Cas-nuclease-P2A-mCherry plasmid with 18 μl of 1 mg/ml
Poly(ethylenimine) (PEI; Sigma). Cas nuclease-transfected cells were
harvested 72 h post-transfection with an Aria SORP flow cytometry
sorter on the basis of mCherry expression. Then, genomic DNA was
extracted for PEM-seq library construction.

### PEM-seq assay and PEM-Q analysis
Each PEM-seq DNA library was constructed according to the standard
procedure[30,38], for which 20 μg of genomic DNA from different Cas
nuclease-edited samples is generally required. The primer control of
each target site was generally applied to the genome of wild-type
HEK293T cells transfected by a Cas nuclease without sgRNA targeting.
For the PEM-seq procedure, first, the genomic DNA was sonicated with
a Covaris M220 Focused Ultrasonicator to obtain 300–700 bp DNA
fragments. Then, a biotinylated primer was designed within 150 bp
from the target site to accomplish primer extension. Biotinylated
single-stranded DNA was enriched with Streptavidin C1 beads and
ligated with a "bridge adaptor", which was designed to achieve expo-
nential amplification of the target fragments. On-bead nested PCR was
performed with I5 and I7 primers followed by size selection and
amplification with indexed Illumina primers. Then, the DNA libraries
were sequenced on an Illumina HiSeq platform by GENEWIZ. The
bioprimers and nested primers used in this study are shown in Sup-
plementary Table 2.

A description of the bioinformatics analysis tools and PEM-Q
pipeline can be found in the previous study[31].

### 293T-EGFP cell GFP disruption assay
The 293T-EGFP reporter cell lines were generated by homologous
repair-mediated gene knock-in. We used a gene target plasmid to cut
the *AAVS1* site and an integration donor plasmid to achieve site-
specific exogenous d2EGFP gene insertion. The EF1-alpha promoter
was used to drive EGFP gene expression. After transducing 293T cells
with the target vector and donor vector, EGFP-positive cells were
selected by FACS for subcloning, and d2EGFP gene integration was
ensured by PCR. For the procedure of editing efficiency detection in
HEK293T-d2EGFP cell lines, plasmids containing a Cas nuclease and a
sgRNA that could target EGFP gene sequences were transfected into

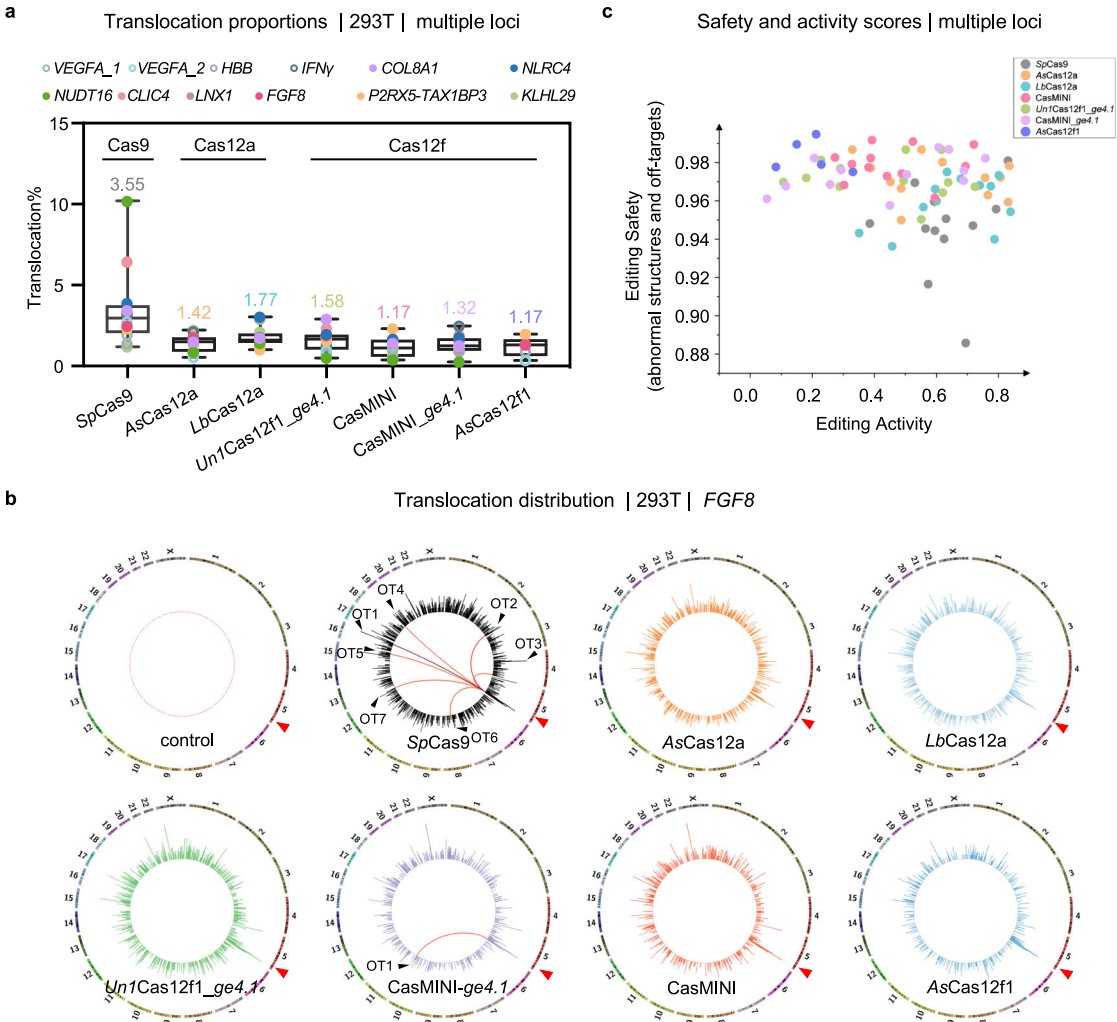

**Fig. 5 | Cas12f nucleases minimized the frequency of chromosomal translocations. a** The percentage of translocations caused by the indicated Cas nucleases on the basis of the total editing events for the indicated 12 loci in HEK293T cells as detected by PEM-seq, with the numbers above the whiskers referring to the average percentages of the translocations at all sites. Values from minimum to maximum are shown by the whiskers, and the bounds of the box indicate the first and third quartile ($N = 12$, for $As$Cas12f1, $n = 5$). The vertical line through the box is the median. Source data are provided as a Source Data file. **b** The translocation distribution patterns across the whole genome (circos plot) for the indicated Cas nucleases at the *FGF18* locus in HEK293T cells. The junction signals were binned into 2-Mb intervals and plotted on the log scale. Red and black arrowheads indicate the on-target and off-target cleavage sites, respectively. **c** Activity and safety scores for all tested Cas nucleases at all 12 loci. Of note, we only could calculate the editing safety scores at effectively cleaved sites, so for $Sp$Cas9, $As$Cas12a, $Lb$Cas12a, $Un1$Cas12f1_ge4.1, CasMINI, and CasMINI_ge4.1, 12 sites were shown; for $As$Cas12f1, five sites were shown. The activity score was referred to actual editing efficiencies of each point; the safety scores were calculated as [1 − (general translocations % + off-target junctions% + large deletions%)].

d2EGFP cells. After the protein was expressed for 1 day, the positively transfected cells were selected with 1.5 µg/µl puromycin for 3 days. The GFP-negative proportion of the cell population was detected by FACS on 5 and 10 days post-transfection and analyzed by FlowJo 10.4 software.

## PEM-Q analysis

Typically, the PEM-Q pipeline can identify several genome editing products: perfect rejoinings, indels, translocations, and other chromosomal abnormalities. The number of indels and translocations to the total number identified products was defined as the editing efficiency ratio. Deletions were defined as small deletions (≤100 bp) and large deletions (>100 bp). Insertions were defined as small insertions (<20 bp) and large insertions (≥20 bp). For off-target analysis, translocation hotspots with sequences very similar to that of the target site (≤8 nt mismatches including both the spacer and PAM sequences) and with more than 3 junctions at the presumed cut-site were considered off-target sites. Additionally, translocation junctions within 100 bp of the detected off-target site were regarded as

off-target translocations. General translocations excluded both junctions within 500 kb upstream and downstream of target sites and off-target translocations.

## Statistics and reproducibility

All biological phenomenon studies were developed with at least five sample sizes. Data are presented as the mean ± SD, the detailed information about sample sizes can be found in figure legends. One-way ANOVA with Geisser-Greenhouse correction statistical analysis was performed on at least three biologically independent experiments by Graphpad prism8, and $p < 0.05$ was considered significant. No statistical method was used to predetermine the sample size. No data were excluded from the analyses, the experiments were not randomized, and the investigators were not blinded to allocation during experiments and outcome assessment.

## Reporting summary

Further information on research design is available in the Nature Research Reporting Summary linked to this article.

## Data availability

The Original PEM-seq sequencing data generated in this study have been deposited in both the NCBI Gene Expression Omnibus (GEO) database under accession code GSE213149 and the NODE (National Omics Data Encyclopedia) database with accession code OEP003371. All plasmids used in this study are available upon request by contacting J.H. (hujz@pku.edu.cn). Except for unforeseen circumstances, requests will be answered within 1 week. Source data are provided with this paper.

## Code availability

The supported PEM-seq analysis code has been uploaded on the GitHub website: https://github.com/JiazhiHuLab/PEM-Q.

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

## Acknowledgements

We thank Dr. Bo Zhang (Peking University) for her generous gift of the *Lb*Cas12a plasmid. We also thank the Flow Cytometry Core at National Center for Protein Sciences at Peking University, particularly Yinghua Guo and Liying Du, for technical help. This work was supported by the Ministry of Agriculture and Rural Affairs of China (NK2022010101), the National Key R&D Program of China (2017YFA0506700), NSFC (grant 32122018 and 31771485), Clinical Medicine Plus X–Young Scholars Project (No. PKU2020LCXQ021), PKU-TSU Center for Life Sciences, and SLS-Qidong Innovation Fund.

## Author contributions

C.X., J.Y., and J.H. conceived and designed the research; M.L. developed the analysis pipeline; C.X., J.Y., S.Y., and L.O. performed the experiments; C.X., J.Y., S.Y., and W.Z. analyzed the data; C.X., J.Y., and J.H. wrote the manuscript.

## Competing interests

The authors declare no competing interests.
