## [Peer Review File · Nature Communications]

Reviewers' Comments:

Reviewer #1:

Remarks to the Author:

Compact type V Cas12 family nucleases can be effectively packed into single AAV for delivery, showing promising potential for in vivo therapeutic applications. Xin et al comprehensively assessed many factors of several compact type V Cas12 nucleases (Cas12f and Cas12e), including the editing efficiencies, offtarget editing activity, and editing patterns. The finding that Cas12f nucleases reduced the levels of chromosomal translocations compared with Cas9 and Cas12a is new, whereas the other findings, such as the editing efficiencies and the editing patterns, have been published previously. Thereby, I think the work is sound, but the novelty is moderate. Here are some specific comments.

1. An engineered Cas12e with drastically enhanced editing efficiency was reported recently (Mol Cell 2022, 82, 1199-1209). The authors need to compare the editing efficiencies of the engineered Cas12e with Cas12f and other large Cas nucleases, which is critical for the readers to select a suitable compact Cas12 nuclease for different applications.
2. The authors used PEM-seq to assess the offtarget editing activities. PEM-seq relies on chromosomal translocations for offtarget editing event detection, which is less frequent. Other methods that directly detect the offtarget editing events, such as guide-seq and digenome-seq, need to be applied to compare the offtarget editing activities of Cas12f, Cas12e with large Cas nucleases.
3. The editing efficiencies of different Cas nucleases were determined by PEM-seq. These results need to be confirmed individually using NGS.

Reviewer #2:

Remarks to the Author:

New miniature CRISPR-Cas nucleases compatible with adeno-associated viral (AAV) delivery have been recently developed as promising tools for human therapeutic applications. However, the systematic study of their efficiency across different genomic sites and safety remained to be established. This timely manuscript aims to bridge this gap by assessing the genome editing efficiency, specificity and off-target activity of miniature nucleases. In this manuscript authors employed high-throughput sequencing assays to evaluate genome editing efficiency, off-target activity and chromosomal translocations resulting due to the double stranded break generated by miniature nucleases. In this benchmarking exercise authors also included Cas9 and Cas12 nucleases enabling direct comparison with miniature nucleases. Although Cas9 and Cas12 still outperform miniature nucleases, further engineering of miniature nucleases may improve their editing efficiency paving the way for their development into robust genome editing tools for human therapeutic applications.

1. What are expression levels of miniature nucleases in HEK cells in comparison to Cas12 and Cas9? Could possible differences in the expression level directly impact genome editing efficiency?
2. The variation of the editing efficiency across different sites for miniature nucleases guided by ge4.1 gRNA is much larger in comparison to the canonical gRNA (Figure 2C). How authors explain it?
3. Can nucleotides flanking the canonical PAM site at different targets impact the editing efficiency across different sites?
4. The authors should also discuss the possibility in the Discussion section, that the observed higher specificity of the Cas12f nucleases might be related to the overall lower activity at the on-target sites in comparison to Cas9 and Cas12a nucleases (Figure 2C);

Minor comments:

- 1) CRISPR/Cas throughout the manuscript should be replaced with CRISPR-Cas;
- 2) p1-2. "Moreover, it can lead to unwanted structural variations in chromatin, including chromosomal translocations, large deletions, and integration of exogenous DNA". The term "chromatin" here and further in the text might be misleading as it assumes that the editing also alters DNA bound proteins.;
- 3) p6-7. "Strikingly, the vast majority of the editing events were deletions by Cas12a and Cas12f,

with percentages greater than 92%, from the lowest for LbCas12a (92.4%) to the highest for AsCas12f1 (96.4%)...". The observed deletions after editing with Cas12 nucleases were also described previously. The authors should add the citations to the relevant literature;

4) The description of Figure 5C is misleading as only the sites that were edited are evaluated (not all twelve loci) (e.g. for AsCas12f1 only five sites are shown).

REVIEWER COMMENTS

Reviewer #1 (Remarks to the Author):

Compact type V Cas12 family nucleases can be effectively packed into single AAV for delivery, showing promising potential for in vivo therapeutic applications. Xin et al comprehensively assessed many factors of several compact type V Cas12 nucleases (Cas12f and Cas12e), including the editing efficiencies, offtarget editing activity, and editing patterns. The finding that Cas12f nucleases reduced the levels of chromosomal translocations compared with Cas9 and Cas12a is new, whereas the other findings, such as the editing efficiencies and the editing patterns, have been published previously. Thereby, I think the work is sound, but the novelty is moderate.

Response: We sincerely thank the Reviewer's comments and valuable suggestions for enhancing the quality of our manuscript. We also thank the recognition of our new findings regarding the editing safety of Cas12f nucleases from the Reviewer. Those comments are all valuable and very helpful in revising and improving our article. We have carefully studied these comments and performed additional experiments or analyses, which we hope to meet with approval. The main corrections in the manuscript and responses to the comments are as follows:

Specific comments:

1. An engineered Cas12e with drastically enhanced editing efficiency was reported recently (Mol Cell 2022, 82, 1199-1209). The authors need to compare the editing efficiencies of the engineered Cas12e with Cas12f and other large Cas nucleases, which is critical for the readers to select a suitable compact Cas12 nuclease for different applications.

Response: We sincerely thank the Reviewer's valuable suggestions. The evaluation of engineered Cas12e could improve the comprehensiveness of our assessment. After sending out the manuscript, we noticed the optimized Cas12e, *PlmCasX-R1-v2* (Tsuchida et al., Mol Cel 2022), and performed a new set of eGFP silencing assay and the PEM-seq analysis in parallel with *PlmCasX* (Response Figure 1). The eGFP silencing assay results showed that *PlmCas12e-R1-v2* dramatically enhanced editing efficiencies at the two tested sites than *PlmCas12e* (46.0% vs. 16.3% and 14.3% vs. 1.8%, respectively). Additionally, the PEM-seq data showed that *PlmCas12e-R1-v2* improved editing efficacies at more than half of the sites. However, only three tested genomic sites had been effectively edited with over 5% efficiencies by *PlmCas12e-R1-v2*, we could not well assess other editing properties such as chromosomal rearrangements and thereby excluded both Cas12e enzymes for further analysis. Since the new set of *PlmCasX* showed highly similar results as the previous set, we

combined the data of *PlmCasX-R1-v2* into Figure 1D and Supplementary Figure 1.

Response Figure 1. Comparison of GFP disruption efficacies between *PlmCas12e* and *PlmCas12e-R1-v2*.

(A) The GFP disruption efficiencies of *PlmCas12e* vs. *PlmCas12e-R1-v2* were shown at the indicated time points and target sites. The GFP disruption proportion is referred to as the number of GFP-negative cells relative to the total number of cells. ‘ng’ represents nontargeting guide RNA. ‘g1’ and ‘g2’ indicate site 1 and site 2, respectively (n=3, the mean ± s.d.).

(B) Representative raw flow cytometry data by *PlmCas12e* and *PlmCas12e-R1-v2* in EGFP disruption assay, with gates showing how the GFP negative cells are gated. (C) Editing efficiency of the control, *PlmCas12e*, and *PlmCas12e-R1-v2* at indicated loci detected by PEM-seq.

2. The authors used PEM-seq to assess the offtarget editing activities. PEM-seq relies on chromosomal translocations for offtarget editing event detection, which is less frequent. Other methods that directly detect the offtarget editing events, such as guide-seq and digenome-seq, need to be applied to compare the offtarget editing activities of Cas12f, Cas12e with large Cas nucleases.

Response: We thank the Reviewer’s valuable comments and concerns. LAM-HTGTS and GUIDE-seq are two of the most sensitive *in vivo* off-target detection methods (Frock et al., Nat Biotechnol 2015; Tsai et al., Nat Biotechnol 2015). Given the

restriction in setting *in vivo* experiments, both methods employed relatively low-frequency events, chromosomal translocations and fragmental integrations, respectively. PEM-seq was developed based on the combination of LAM-HTGTS and targeted sequencing (Yin et al., Cell Discovery 2019; Liu et al., Nucleic Acids Res 2021). With regards to off-target detection, PEM-seq shows similar sensitivity as LAM-HTGTS and GUIDE-seq in previous publications (Response Table 1). Our previous study detected 18 off-targets for the *EMX1* site by *SpCas9* via PEM-seq, with a loss of 3 weak off-targets and a gain of 5 new off-targets compared to GUIDE-seq analysis (Yin et al., Cell Discovery 2019, Supplementary Fig. S3A). Also, another group applied GUIDE-seq and PEM-seq to evaluate the off-target effects of CRISPR-cCas12a, and both methods consistently detected a weak off-target of the human *DNMT1* locus in HEK293T cells (Ling et al., Mol Cell 2021, Figure 3F). Therefore, we hope the Reviewer will agree with us that either PEM-seq or GUIDE-seq is suitable for *in vivo* off-target detection.

The *in vitro* off-target detection methods such as Digenome-seq, CIRCLE-seq, and SITE-seq always showed more off-target sites than the *in vivo* method, but the varied accessibility at these off-target sites lead to inconsistency with the *in vivo* methods.

We hope that our detailed clarifications and explanations have addressed the Reviewer's concerns.

Research	Target sites	Cell type	HTGTS	PEM-seq	GUIDE-seq	Digenome-seq
Yin et al., Cell Discovery 2019	EMX1	U2OS	—	18 OTs	15 OTs	—
Ling et al., Mol Cell 2021	DNMT1	HEK293T	—	1 OT	1 OT	—
Liu et al., Nucleic Acids Res 2021	RAG1	HEK293T	33 OTs	59 OTs	—	—
Dobbs et al., Nat Commun 2022	EMX1	HEK293T	12 OTs	—	13 OTs	31 OTs
Frock et al., Nat Biotechnol 2014	VEGFA	HEK293T	38 OTs	—	21 OTs	—
Tsai et al., Nat Biotechnol 2015		U2OS				

Response Table 1. The Off-target numbers are assessed by various methods.

3. The editing efficiencies of different Cas nucleases were determined by PEM-seq. These results need to be confirmed individually using NGS.

Response: We are grateful for the Reviewer's kind recommendations. PEM-seq is a high-throughput sequencing method for evaluating genome editing outcomes in a combination of LAM-HTGTS and targeted sequencing as abovementioned. During the library preparation, the primer extension step in PEM-seq captures the genomic fragments containing the target sites. Therefore, PEM-seq contains all the information targeted sequencing has, but PEM-seq employs a unique molecular index (UMI) that is lacking in amplicon-based targeted sequencing, which makes PEM-seq more

accurate in quantification. As suggested, we used amplicon-based targeted sequencing to confirm the editing efficiencies of 3 target sites within or adjacent to *COL8A1*, *FGF18*, and *P2RX5-TAX1BP3* genes. We used the CRISPResso2 pipeline to analyze the sequencing data (Response Figure 2A) (Clement et al., Nat Biotechnol 2019). We found that the relative editing efficiencies of different Cas nucleases identified by these two assays were generally consistent. Notably, since targeted sequencing also counts substitutions as editing events, editing efficiencies assessed by targeted sequencing were slightly higher than that of PEM-seq (Response Figure 2B).

Response Figure 2. Gene editing efficiencies of different Cas nucleases assessed by PEM-seq and Amplicon-seq

(A) The design and experimental procedure of Amplicon-seq. (B) The gene editing efficiencies detected by PEM-seq and Amplicon-seq in *COL8A1*, *FGF18*, and *P2RX5-TAX1BP3* target sites, n=1. Of note, the last gray bars represent data from the control samples which have no editing.

Reviewer #2 (Remarks to the Author):

New miniature CRISPR-Cas nucleases compatible with adeno-associated viral (AAV) delivery have been recently developed as promising tools for human therapeutic applications. However, the systematic study of their efficiency across different genomic sites and safety remained to be established. This timely manuscript aims to bridge this gap by assessing the genome editing efficiency, specificity and off-target activity of miniature nucleases. In this manuscript authors employed high-throughput sequencing assays to evaluate genome editing efficiency, off-target activity and chromosomal translocations resulting due to the double stranded break generated by miniature nucleases. In this benchmarking exercise authors also included Cas9 and Cas12 nucleases enabling direct comparison with miniature nucleases. Although Cas9 and Cas12 still outperform miniature nucleases, further engineering of miniature nucleases may improve their editing efficiency paving the way for their development into robust genome editing tools for human therapeutic applications.

Response: We are very grateful for the Reviewer's support of our manuscript and insightful suggestions. We also thank the Reviewer's comments that our manuscript timely bridged the gap in the miniature nucleases' editing properties. And other constructive comments are all valuable for improving our paper. We have carefully studied the comments and have made revisions and corrections to the article. Below please find specific responses to the reviewer's remaining concerns, which we believe have improved the quality of the work.

Specific comments:

1. What are expression levels of miniature nucleases in HEK cells in comparison to Cas12 and Cas9? Could possible differences in the expression level directly impact genome editing efficiency?

Response: We thank the reviewer's constructive suggestions and questions. We introduced an N-terminal FLAG tag at Cas nucleases in the expression vectors, then collected transfected HEK293T cells by fluorescence-activated cell sorting (FACS) based on mCherry fluorescence 72h post-transfection as done previously. Whole Cell Extract (WCE) were then isolated to prepare the western blot samples. WCE from 40,000 sorted cells was used for each sample, and housekeeping protein Histone H3 was chosen as a loading control. Western blot results showed that the relative expression levels of Cas12a (*AsCas12a*, *LbCas12a*) and Cas12f (*CasMINI*, *CasMINI_ge4.1*, *Un1Cas12f1_ge4.1*) were generally consistent, and the expression levels of Cas9 were significantly lower than those of Cas12 family (Response Figure 3), suggesting that the expression level is not a key factor for gene editing in this

study.

Response Figure 3. Western blot showing the FLAG-tagged Cas nucleases expression level.

2. The variation of the editing efficiency across different sites for miniature nucleases guided by ge4.1 gRNA is much larger in comparison to the canonical gRNA (Figure 2C). How authors explain it?

Response: We thank the reviewer's constructive questions. We also notice this phenomenon but have no good explanation currently. The ge4.1 gRNA modified five parts of the canonical gRNA of *UnrCas12f1*: corrected an internal penta(uridinylate) (UUUUU) sequence; added 5'-U4RU4 (R = A or G) sequence to the 3' terminus of crRNA; truncated the 5' termini of the trans-activating CRISPR RNA (tracrRNA); trimmed the entire tracrRNA–crRNA complementary region; truncated the disordered stem 2 region in the tracrRNA. *Cas12f_ge4.1* truncated the tracrRNA stem 2 region as the segment from A⁻¹²⁹ to U⁻¹⁰³ was reported disordered (Takeda et al., Mol Cell 2021). Those changes have impacts on the binding stability of Cas12 and target DNA, which may lead to varying levels of changes in editing efficiency at different sites by miniature nucleases guided by ge4.1 gRNA. We agree with the Reviewer that this is an interesting question and is worth further exploring in the future.

3. Can nucleotides flanking the canonical PAM site at different targets impact the editing efficiency across different sites?

Response: We sincerely thank the Reviewer's constructive suggestions and questions. We analyzed our data to investigate the effect of nucleotides flanking the PAM, but the relatively few tested sites make it difficult to draw a solid conclusion. We firstly examined the gRNA–distal nucleotide adjacent to the PAM as presented in Response Figure 4A. The A nucleotide ensures consistent high editing efficiency at the tested sites for both *SpCas9* and most Cas12 enzymes. The T nucleotide also shows high editing efficiency for *SpCas9* but not for the Cas12 family. We then examined the

gRNA-proximal nucleotide adjacent to the PAM as presented in Response Figure 4B. T nucleotide is unfavorable for *SpCas9* at tested sites, in line with previous findings (Doench et al., Nat Biotechnol 2014; Xu et al., Genome Res 2015). While no consistent conclusion can be drawn about Cas12 due to limited sites.

Response Figure 4. Gene editing frequencies of different nucleotides flanking the PAM.

(A) Gene editing frequencies of four different nucleotides flanking the PAM at the genome end following edited by *SpCas9*, *AsCas12a*, *LbCas12a*, *CasMINI*, *CasMINI_ge4.1*, *Un1Cas12f1_ge4.1*, and *AsCas12f1*. (B) Gene editing frequencies of four different nucleotides flanking the PAM at the gRNA-end following edited by *SpCas9*, *AsCas12a*, *LbCas12a*, *CasMINI*, *CasMINI_ge4.1*, *Un1Cas12f1_ge4.1*, and *AsCas12f1*.

4. The authors should also discuss the possibility in the Discussion section, that the observed higher specificity of the Cas12f nucleases might be related to the overall lower activity at the on-target sites in comparison to Cas9 and Cas12a nucleases (Figure 2C);

Response: We appreciate the Reviewer’s insightful comments and suggestions. We have revised the discussions section in our revised manuscript as “The observed higher specificity of the Cas12f nucleases might be due to the overall lower activity at the on-target sites in comparison to Cas9 and Cas12a nucleases. However, we also noticed that both *CasMINI* and *AsCas12f1* had undetectable off-target effects at the

tested sites and generated very few large deletions or translocations in some effectively edited loci (Figure 2D, Figure 3E, Figure 5A and Supplementary Figure S3B), which suggested that the long over-hangs (~11bp) of cleaved ends may be also involved in suppressing structural variations by affecting the DNA repair pathways.”

Minor comments:

1) CRISPR/Cas throughout the manuscript should be replaced with CRISPR-Cas;

Response: We sincerely thank the Reviewer’s kind suggestions. We have adjusted all the CRISPR/Cas mentioned in the manuscript to be CRISPR-Cas.

2) p1-2. “Moreover, it can lead to unwanted structural variations in chromatin, including chromosomal translocations, large deletions, and integration of exogenous DNA”. The term “chromatin” here and further in the text might be misleading as it assumes that the editing also alters DNA bound proteins.;

Response: We are grateful for the Reviewer’s kind comments and sorry for this confusion. We have corrected the term “chromatin” to be “chromosome” in the manuscript and changed other “chromatin structural variations” into “structural variations” as suggested.

3) p6-7. “Strikingly, the vast majority of the editing events were deletions by Cas12a and Cas12f, with percentages greater than 92%, from the lowest for LbCas12a (92.4%) to the highest for AsCas12f1 (96.4%)...”. The observed deletions after editing with Cas12 nucleases were also described previously. The authors should add the citations to the relevant literature;

Response: We sincerely thank the Reviewer’s kind comments and literature suggestions. We have supplemented relevant literature citations of deletions-associated descriptions in our manuscript (Wu et al., Nat Chem Bio 2021; Kim et al., Nat Biotechnol 2022).

4) The description of Figure 5C is misleading as only the sites that were edited are evaluated (not all twelve loci) (e.g. for AsCas12f1 only five sites are shown).

Response: We thank the Reviewer’s comments, and we are sorry for this confusion. For the sites with very low editing levels, it’s difficult to calculate the safety score, so we have to discard those sites. We have adjusted the description of Figure 5C to “we calculated the editing safety score of these enzymes by combining their off-target activity with the extent of structural variations they induced at effectively edited sites and aligned the score on the basis of the editing efficiency score.” Also, we have added the detailed statement into the legend of Figure 5C: “Of note, we only could calculate the editing safety scores at effectively cleaved sites, so for *SpCas9*,

AsCas12a, *LbCas12a*, *UnlCas12f1_ge4.1*, CasMINI, and CasMINI_*ge4.1*, twelve sites were shown; for *AsCas12f1*, five sites were shown. The activity score was referred to actual editing efficiencies of each point; the safety scores were calculated as $[1 - (\text{general translocations}\% + \text{off-target junctions}\% + \text{large deletions}\%)]$.” We hope that the revised manuscript is now clear enough to convey detailed information.

Reviewers' Comments:

Reviewer #1:

Remarks to the Author:

The authors have fully addressed my previous concerns.

Reviewer #2:

Remarks to the Author:

Authors addressed most of the questions raised in my previous review by performing additional experiments or providing reasonable explanations clarifying points raised in the review.